# The Role of Imaging in Portal Vein Thrombosis: From the Diagnosis to the Interventional Radiological Management

**DOI:** 10.3390/diagnostics12112628

**Published:** 2022-10-30

**Authors:** Paolo Marra, Ludovico Dulcetta, Francesco Saverio Carbone, Riccardo Muglia, Giuseppe Muscogiuri, Maurizio Cheli, Lorenzo D’Antiga, Michele Colledan, Stefano Fagiuoli, Sandro Sironi

**Affiliations:** 1Department of Radiology, ASST Papa Giovanni XXIII Hospital, 24127 Bergamo, Italy; 2School of Medicine and Surgery, University of Milano-Bicocca, 20126 Milan, Italy; 3Department of Radiology, IRCCS Istituto Auxologico Italiano, San Luca Hospital, 20149 Milan, Italy; 4Department of Pediatric Surgery, ASST Papa Giovanni XXIII Hospital, 24127 Bergamo, Italy; 5Department of Pediatric Hepatology, Gastroenterology and Transplantation, ASST Papa Giovanni XXIII Hospital, 24127 Bergamo, Italy; 6Department of Organ Failure and Transplantation, ASST Papa Giovanni XXIII Hospital, 24127 Bergamo, Italy; 7Department of Gastroenterology, Hepatology and Transplantation Unit, ASST Papa Giovanni XXIII Hospital, 24127 Bergamo, Italy

**Keywords:** portal vein thrombosis, pediatric liver transplantation, interventional radiology

## Abstract

Purpose: To illustrate diagnostic and interventional imaging for the characterization and treatment of portal vein thrombosis (PVT). Introduction: The broad spectrum of congenital and acquired PVT manifestations is illustrated, with a focus on the pediatric population; diagnostic and interventional imaging techniques are described. Description: PVT frequently presents as an incidental finding at imaging in the screening for liver diseases or for other unrelated reasons. PVT can be classified based on: extension (intrahepatic, extrahepatic, involving the spleno-mesenteric tract, etc.); degree (partial or complete); onset (acute or chronic); and with or without cavernomatous transformation. This comprehensive review relies on the experience gained from a large series of congenital and acquired PVT in a referral center for pediatric and adult liver transplantation. Diagnostic and interventional imaging techniques are described, including: color–Doppler and contrast-enhanced Ultrasound; CT and MR angiography; retrograde portography; percutaneous transhepatic, transplenic, and transmesenteric portography; transjugular intrahepatic portosystemic shunt creation. Pre- and post-operative imaging assessment of the surgical meso-rex bypass is discussed. The description is enriched with an original series of pictorial imaging findings. Conclusion: PVT is a clinical condition associated with significant morbidity and mortality. Diagnostic and interventional imaging plays a crucial role in both conservative and operative management.

## 1. Introduction

The term portal vein thrombosis (PVT) refers to the complete or partial obstruction of blood flow in the portal vein, due to the presence of a thrombus in the vasal lumen [1]. The underlying etiology of PVT remains unclear in up to 50% of children and adults [2]. PVT, either primary or secondary to an underlying chronic liver disease, is the most common cause of prehepatic portal hypertension. On the one hand, primary PVT can be idiopathic, iatrogenic (neonatal umbilical vein catheterization), or related to an altered coagulation status. On the other hand, the progression of most chronic liver diseases leads to fibrosis, which increases sinusoid resistance and determines an abnormal elevation of portal venous pressure. The spontaneous development of a collateral circulation to divert the flow to the systemic circulation is the first observed anomaly, which itself contributes to PVT. 

Liver transplantation (LT) is the standard of care for adult and pediatric patients with end-stage liver disease and liver-based metabolic disorders, either acute or chronic. Interestingly, up to 50% of patients with PVT at the time of LT are previously unrecognized [3,4]. In this setting, imaging plays a crucial role in a timely diagnosis of PVT to prevent an irremediable progression. PVT can also occur as a complication of LT, and it is associated with high mortality and graft loss. Although the incidence is very low in adults [5], PVT leads to a reduction in 5-year graft survival when compared with liver transplant recipients without portal vein complications [6]. Compared to adults, pediatrics are at greater risk of developing post-transplant PVT with an incidence rate raising up to 3–14% after a living-donor LT compared with 2–3% after a deceased-donor LT [7]. Many causes may lead to PVT in the setting of post-LT: on the one hand immediate vascular thrombosis and the portal flow-steal phenomenon in early PVT; on the other hand, portal vein stenosis in chronic PVT. 

PVT complications ordinarily require percutaneous recanalization by means of different interventional radiological approaches or repeat surgery and can lead to retransplantation and/or recipient death [8].

In this review, we discuss the features of PVT with insights into diagnostic issues, providing an overview of the typical and unusual findings observed by multimodality imaging.

## 2. Etiology

In population studies, non-malignant PVT can be encountered in the following scenarios (Table 1): acute and chronic non-cirrhotic liver diseases or malformations and perinatal portal venous system injuryend-stage chronic liver diseasesand complications of LT [2,9,10,11].

**Table 1 diagnostics-12-02628-t001:** Etiopathogenesis of non-malignant portal vein thrombosis.

Non-Cirrhotic Systemic Causes	
Congenital Factor V Leiden mutation Factor II (prothrombin) mutation Protein C deficiency Protein S deficiency Prothrombine mutation Antithrombin deficiency	Acquired Myeloproliferative disorders Antiphospholipid syndrome Paroxysmal nocturnal hemoglobinuria Oral contraceptives Pregnancy Hyperhomocysteinemia
Non-cirrhotic local causes	
Congenital Portal vein stenosis Portal vein atresia Portal vein agenesis	Acquired Neonatal omphalitis Neonatal peritonitis Umbilical vein catheterization Diverticulitis, appendicitis Pancreatitis Duodenal ulcer Cholecystitis Tuberculous lymphadenitis Crohn’s disease, ulcerative colitis Cytomegalovirus hepatitis Liver transplantation Splenectomy GI surgery Abdominal trauma Surgical portosystemic shunting, transjugular intrahepatic portosystemic shunt, Iatrogenic
Cirrhotic-related diseases	
Genetic/Congenital Alagille syndrome Biliary atresia Primary biliary cirrhosis Primary sclerosing cholangitis Alpha-1 antitrypsin deficiency Wilson disease Hemochromatosis Nonalcoholic fatty liver disease Autoimmune hepatitis Cystic fibrosis	Acquired Chronic alcohol abuse Chronic viral hepatitis Chronic hepatotoxic drugs Schistosomiasis Sarcoidosis Cryptogenic cirrhosis Metabolic disorders

### 2.1. Acute and Chronic Non-Cirrhotic Liver Diseases

The non-cirrhotic causes of PVT can be subdivided into three main categories of congenital and acquired conditions: portal vein damage and subsequent obstruction such as omphalitis, umbilical vein catheterization, neonatal peritonitis, abdominal trauma, and iatrogenic operative trauma to the portal vein;congenital anomalies such as portal vein stenosis, portal vein atresia, or agenesis. Obstruction can occur anywhere along the line of left and right vitelline veins from which the portal vein develops. In these cases, PVT is often associated with the presence of other congenital defects, usually of the cardiovascular system;factors indirectly associated with PVT, such as neonatal systemic sepsis from non-intraabdominal sources, dehydration, multiple exchange transfusions, and hypercoagulable states, including myeloproliferative disorders such as polycythemia vera, inherited deficiencies of natural anticoagulants such as antithrombin III, protein C and protein S, activated protein C resistance (APCR), and prothrombin gene (G20210A) mutation [2,12].

### 2.2. End-Stage Chronic Liver Diseases

Progression of most chronic liver diseases leads to fibrosis, cirrhosis, and consequently to an increase in portal venous pressure. The portal vein flow is compromised by two factors:the increase of resistance caused by parenchymal architectural distortion leads to increased vascular resistance, decreased flow, and in some cases to the inversion of the flow, and eventually to portal obstruction. A blood flow slower than 15 cm/s in the portal vein is associated with an increased risk of PVT [13];the development of hepatofugal portosystemic collateral vessels due to fibrosis and high hepatic sinusoidal pressure. Decreased portal vein inflow caused by portosystemic collateral vessel shunting may result in PVT [14];

Moreover, it has been suggested that decreased portal blood flow and the presence of periportal lymphangitis and fibrosis promote thrombus formation [2]. 

A leading cause of PVT in advanced cirrhosis might be the simultaneous presence of the three components of Virchow’s triad: venous stasis, endothelial injury, and hypercoagulopathy [15]. Of note, these pathogenetic mechanisms are frequently observed in young children with biliary atresia [16].

### 2.3. PVT after Liver Transplantation

The patency of vascular reconstructions is an important factor in the success of LT. Adequate PV inflow is mandatory for rapid regeneration of transplanted liver during the immediate intra- and postoperative phase, particularly in pediatric LT recipients with split-liver grafts [6,17]. Although recent improvements in surgical techniques have contributed to decrease vascular complications after LT, PV complications are still a significant cause of postoperative graft failure and morbidity, occurring in 1–2% of adult deceased-donor LT and in 3–19% of pediatric LT [8]. In particular, PVT is a rare but serious complication following LT, with a reported incidence of 1.16–2.7% [18,19].

The incidence of portal vein complications is usually higher in pediatric recipients than in adults, due to the more common use of segmental liver grafts [20].

Risk factors associated with PVT include young age, body weight < 6 kg, technical problems (modification of the standard end-to-end anastomosis of the portal vein due to preexisting PVT, and use of venous grafts for portal vein reconstruction), graft torsion, splenectomy during transplantation, small portal vein diameter, size mismatch between donor’s and recipient’s portal vein, and short pedicle of the donor’s portal vein [11,18,19,20,21,22]. Longer cold ischemia time (>12 h) can also be a risk factor for developing venous problems [11]. In patients affected by chronic liver disease, the liver’s recovery from cold ischemia is compromised by the presence of large varices responsible for the extensive routing of blood out of the portal system, representing a cause of PVT; indeed, in situations in which intrahepatic vascular resistance to venous flow remains elevated after LT, significant hepatofugal portal flow may continue through persistent portosystemic collateral shunts. This condition—the so-called portal flow-steal phenomenon - can result in early PVT [16,23]. Especially in children, several reports have demonstrated that the diversion of portal vein flow from segmental liver grafts may be life-threatening, indicating that large collateral vessels should be interrupted [16,24].

Additionally, biliary atresia, the leading indication for transplantation in children, is associated with portal vein sclerosis and hypoplasia, representing another technical issue during vascular reconstruction [10,25]. Recently, in a large series of children following LT, it was demonstrated that patients receiving a segmental graft (living-related, reduced size, or split-liver), had a higher risk of PVT than those receiving a whole liver. This difference was found both in the short-term (30 days) and long-term (24 months), follow-up [26].

Finally, PVT can be sustained by severe forms of anatomical defects, such as portal vein stenosis or anastomotic kinking. These conditions can be asymptomatic, with or without abnormal liver function tests, often representing an incidental finding during follow-up of liver-transplanted patients [27].

Possible consequences of PVT include graft dysfunction, graft loss, portal hypertension, limitations of the quality of life, recurrent thrombosis, symptomatic portal biliopathy, and death; moreover, in pediatric LT recipients PVT is associated with growth retardation and neurocognitive impairment [8,17]. Graft loss appears to be particularly associated with complete PVT in the early post-transplant course [28].

In patients with long-term PVT after LT, ischemic damage to the hepatic parenchyma and ischemic biliopathy may cause biliary cirrhosis of the graft, even though cavernomatous formation of the portal vein developed around the obstructed portal vein. Even when liver grafts are free from severe fibrosis, portosystemic shunt may cause an intrapulmonary shunt or pulmonary hypertension. To avoid graft loss from ischemia-induced biliary cirrhosis or to avoid the severe complication of hepatopulmonary syndrome requiring retransplantation, early detection of PVT is fundamental [21].

## 3. Classification

PVT can be evaluated on its:extension (intrahepatic, extrahepatic, or both; mono- or polidistrectual);degree (partial or complete);and onset (acute or chronic).

### 3.1. Extension

To correctly classify PVT, it requires assessing the number of vessels of the splanchnic venous system that are involved. Several classifications of PVT have been proposed. However, the most widely used is the one by Yerdel et al. that has been demonstrated to have a prognostic value in patients who are submitted to LT:grade I, thrombus at main portal vein affecting less than 50% of the lumen with or without minimal extension into the superior mesenteric vein (SMV);grade II, thrombus at portal vein affecting more than 50%, including complete thrombosis, with or without minimal extension into the SMV;grade III, complete PVT plus thrombosis extending into the proximal SMV with patent distal SMV;grade IV, complete PVT plus complete thrombosis of the SMV [3,29,30].

Recently, Bhangui et al. have proposed a dichotomy for PVT, which includes complex PVT and noncomplex PVT. This classification has been used as a guide for portal flow reconstruction during LT [29,30,31,32,33]. 

From a radiological perspective, PVT can be classified into four categories, depending on the involvement of: intrahepatic portal vein;extrahepatic portal vein;both intrahepatic and extrahepatic portal veins;SMV and/or SV.

To our knowledge, no classification has focused on the different involvement of the splenic and mesenteric venous system that, indeed, is relevant, since it is potentially associated with the development of variable complications, which require specific treatment approaches.

The broad spectrum of clinical manifestations of PVT is highly related to the affected site. The occlusion of the extrahepatic portal and splenic vein can cause complications mainly related to portal hypertension. When mesenteric vein obstruction is present, intestinal ischemia and eventually infarction (with a high rate of complications and mortality) can occur because of complete obstruction of venous outflow [28,34].

### 3.2. Degree

In the classification of PVT, the percentage of lumen occlusion is crucial as it may impact the treatment approach. PVT has a wide spectrum of severity, from partial occlusion to cavernous transformation. Differentiation between partial and complete PVT can be made using the following criteria: partial PVT is defined as the presence of a residual lumen in the vein, with a peripheral circulating channel; complete PVT is defined as a completely obstructed vein [35]. Chronic complete PVT evolves into a cavernous transformation that entails the disappearance of the native portal vein, replaced by tortuous peri-hilar vessels.

The degree of occlusion correlates with a progressive severity of the clinical manifestation of PVT. Complete obstruction can lead to hepatic decompensation, variceal bleeding and intestinal infarction [15].

### 3.3. Onset

From a clinical point of view, PVT consists of two different entities, acute PVT, and chronic PVT, which represent subsequent stages of the same disease and share similar causes but differ in their management [36]. Establishing the onset timing of thrombosis (acute vs. chronic) is central to estimating prognosis and to aid therapeutic decision-making [37].

Acute PVT is defined as a sudden or recent formation of a thrombus within the portal vein. The thrombus can involve the mesenteric and/or splenic veins; occlusion may be complete or partial [28,38]. Acute PVT has been rarely reported in children [28].

If not resolved, acute and complete PVT can evolve into chronic PVT, leading to the cavernous transformation of the portal vein. This frequently leads to massive esophageal, gastric, gastroesophageal, or ectopic varices and potentially bleeding. Additionally, patients can have refractory ascites and/or hydrothorax [39,40].

In the absence of cirrhosis or other concomitant causes of portal hypertension, signs of severe portal hypertension, such as ascites, splenomegaly, and portosystemic collaterals, also suggest chronicity of PVT [38].

In an LT setting, PVT onset can be either classified as early (within a month after LT) or late (more than a month after LT). The majority of PVTs (>80%) are diagnosed within a month (early PVT) from LT and are mostly due to suboptimal grafts, hypercoagulable states, portal venous stasis, and possibly underlying anatomical defects. On the other hand, late PVT may actually be misdiagnosed early thromboses due to portal vein stenosis and/or kinks [27].

#### 3.3.1. Cavernous Transformation

Following acute thrombosis without spontaneous or medical recanalization, the portal venous lumen obliterates, and porto-portal collaterals develop. The main portal venous trunk may shrink and be replaced by multiple tortuous vessels from both the portoportal and portosystemic collaterals, bypassing the thrombosed portion of the portal vein. This process is called cavernous transformation, the result of which is the portal cavernoma, which fully develops in as few as 6 days to a couple of months after acute thrombosis [38].

A diagnosis of cavernoma is readily made by abdominal imaging with ultrasound, computed tomography (CT), or magnetic resonance (MR) which show serpiginous structures in place of the main portal vein and/or its main branches [40,41]. In some patients, a large, prominent collateral vein at the porta hepatis can be mistaken for a normal portal vein [28] (Figure 1). A study by Marin et al. demonstrated that patients with portal cavernoma may also develop focal nodular hyperplasia (FNH)-like lesions in up to 21% of cases. The development of these benign tumors is thought to be related to an increased arterial perfusion of the liver parenchyma, to compensate the reduced portal flow [42].

#### 3.3.2. Portal Biliopathy

Portal biliopathy is defined as the presence of abnormalities of the extrahepatic and intrahepatic bile ducts and gallbladder walls in patients with noncirrhotic/non-neoplastic PVT [2] (Figure 2).

Although radiologic manifestations of portal cavernoma biliary damage are described in up to 100% of patients with extrahepatic PVT, most of them are initially asymptomatic. In one study [43], only 1 of 20 patients had choledocholithiasis, whereas, in another study [44], biliary changes were associated mainly with biochemical abnormalities.

However, progressive portal cavernoma cholangiopathy may become symptomatic in 5% to 38% of patients presenting with jaundice, pain, recurrent cholangitis, and cholecystitis, typically 8–10 years after diagnosis, and leads to severe complications, including secondary biliary cirrhosis in the case of prolonged bile duct obstruction [45,46].

Biliary abnormalities are common in PVT because paracholedochal and paracholecystic veins form the portoportal collaterals to bypass the obstructed segment of the portal vein. Biliary abnormalities may be explained either by compression of bile ducts by prominent paracholedochal and epicholedochal collateral vessels (indentations and wall irregularities) or ischemic injury of the bile ducts as a result of thrombosis of veins draining the bile duct (leading to ischemic stricture development). Biliary stricture is a frequent finding on MR examinations in the setting of portal cavernoma biliopathy, affecting up to 93.7% of patients [47].

Alterations in portal biliopathy can be assessed by endoscopic retrograde cholangiopancreatography (ERCP), CT, or MR examinations with the following proposed classification [48], based on the involvement of: type I: only extrahepatic bile duct;type II: only intrahepatic bile duct;type IIIa: extrahepatic and unilateral (right or left) intrahepatic bile duct;type IIIb: extrahepatic and bilateral intrahepatic ducts.

## 4. Role of Non-Invasive Imaging Modalities in Portal Vein Thrombosis

### 4.1. Color-Doppler Ultrasonography (CDUS) and Contrast-Enhanced Ultrasound (CEUS)

Color-Doppler Ultrasonography (CDUS) is the first-line diagnostic approach when PVT is suspected. It is non-invasive, low cost, bedside, repeatable, and does not expose the patient to radiation. The diagnostic sensitivity and specificity for CDUS range from 66% to 100% depending on the expertise of the individual examiner and the extent of PVT [3]. CDUS is commonly used in the initial diagnosis and follow-up of patients with suspected or known PVT as it readily depicts occlusion of the portal vein and the presence of cavernous transformation with a sensitivity and specificity of approximately 95% [49].

Compared to further imaging modalities, CDUS has shown to be useful for an accurate assessment of the portal venous system. Extrahepatic visceral vessels leading to portal cavernoma as well as collaterals caused by portal hypertension can be detected in a dynamic mode and with good tissue–vessel discrimination [50].

The ultrasonographic diagnosis of PVT relies on the presence of echogenic material obstructing the lumen of the vessel and the complete or partial absence of flow in the portal vein at color flow Doppler imaging [51] (Figure 3). As the chronic thrombus ages and calcifies, CDUS may show areas of increased echogenicity. In addition, the collateral veins from cavernous transformation around the segments of PVT display a serpiginous mixed-color pattern on CDUS and are anechoic on gray-scale imaging [39,52]. The appearance of the portal cavernoma has been described as a “spongelike mass”, a distinctive tangle of tortuous vessels in the porta hepatis characterized by hepatopetal flow with loss of normal portal phasicity [2] (Figure 2). However, discriminating cavernous collaterals from the portal vein in children with PVT can be very difficult.

One study [53] showed that CDUS can play an important role in the diagnosis of PVT by evaluating indices of arterial resistance at the hepatic and splenic levels. PVT produces a buffer response that involves vasodilatation of the hepatic artery to maintain adequate hepatic perfusion. In the acute and subacute phases, this phenomenon can be documented with CDUS as decreased hepatic artery resistance. Moreover, PVT may be associated with increased arterial resistance indices at the splenic level because of the obstruction to splenic outflow [53].

CDUS is useful in assessing the patency of the intrahepatic portal veins as well as the splenic and superior mesenteric veins, which may also be occluded in PVT, but is less reliable in determining the extension of the thrombus to the mesenteric circulation [49,54].

In portal biliopathy CDUS provides additional information regarding the presence of gallbladder varices, observed in about 35% of patients. Gallbladder varices appear as tortuous, dilated vessels in or around the wall of the gallbladder or in the gallbladder fossa. Moreover, ultrasound can detect the presence of cholelithiasis caused by biliary obstruction secondary to portal biliopathy: gallstones are characteristically echogenic and demonstrate posterior acoustic shadowing regardless of the gallstone composition [48] (Figure 2).

Limitations of CDUS include the patient’s body habitus and intestinal gas interfering with the examination, specifically when evaluating PVT extension into the mesenteric and splenic veins. Additionally, ultrasound is operator-dependent, requiring an experienced sonographer [15].

Contrast-enhanced ultrasound (CEUS) has been proposed to overcome the limitations of CDUS by improving visualization of portal vein vasculature, especially in post-LT follow-up [55]. The European Federation of Societies for Ultrasound in Medicine and Biology (EFSUMB) published an update in 2012 which recommended CEUS for the confirmation of occlusion of the portal vein [56].

CEUS typically shows the absence of portal vein enhancement within the thrombosed segment. Distal to the thrombosed segment, there may be enhancement of the intrahepatic portal vein branches, due to a phenomenon called “arterial backflow”. This is secondary to the physiological connections between the hepatic artery and portal vein branches at the levels of sinusoids, vasa vasorum of the portal vein, or peribiliary vascular plexuses [57].

Despite the advantages of CEUS examination such as the non-nephrotoxicity of the ultrasound contrast agents (unlike the contrast agents used in CT and MR), the wide availability, the lesser expensiveness than other imaging methods, CEUS is still not commonly used in post-LT work-up examinations and literature on its use for the evaluation of PVT is scarce. This is probably because CEUS presents the same limitation of general Ultrasound, in particular poor standardization, and patient’s condition/operator’s skillness dependence [55].

### 4.2. CT and MR

CT and MR can be used to confirm the presence of chronic PVT. The thrombosed vessel may be obliterated, or it may appear attenuated with the incorporation of a clot into the wall. In the chronic form, the thrombosed vessel can be hypodense, containing linear areas of calcification within the thrombus. In addition, CT and MR can depict features of portal hypertension, including ascites, splenomegaly, and portosystemic shunts [39,58].

Advantages of CT and MR over CDUS include the possibility of detecting the wide spectrum of PVT features, such as bowel ischemia, septic foci, intraabdominal malignancies, and higher sensitivity in the detection of thrombosis in the splenic vein and SMV. 

#### 4.2.1. CT

CT provides a wider field of view and is less operator dependent than CDUS. Many of the findings are easily identified on portal venous phase images, though multiphase imaging with non-contrast, arterial, portal venous, and potentially delayed phases may optimize the information obtained from the examination [49]. 

A CT scan in the pre-contrast phase can show hyperattenuating material in the portal vein. After contrast injection, lack of luminal enhancement increased portal vein caliber, increased arterial hepatic enhancement, and decreased portal hepatic enhancement are typical findings of PVT [28] (Figure 3 and Figure 4). Transient hepatic attenuation differences related to altered hepatic perfusion can also be demonstrated on CT with geographic areas of increased attenuation at the periphery of the liver on arterial phase imaging. These transient attenuation differences are related to decreased portal venous perfusion and compensatory increased arterial perfusion from the periphery of the liver [46,49]. 

In the chronic form, the thrombosed vessel can be hypodense, containing linear calcifications, obliterated or shrunk, with cavernous transformation [39,58]. In candidates for LT, calcifications in the portal vein should be actively investigated, because they indicate a more fragile vessel and can hinder surgical anastomoses creation [58]. Splenomegaly and splenic artery aneurysms are also relevant findings readily depicted by CT (Figure 5).

CT provides a better assessment of thrombus extension within the portal venous system than CDUS, because of its higher sensitivity and accuracy than CDUS for identifying thrombosis of the SMV and splenic vein [35]. If the SMV is involved by thrombosis, thinning of the intestinal wall or lack of mucosal enhancement of a thickened intestinal wall after contrast injection are further pieces of evidence for intestinal infarction [28] (Figure 6).

Enlargement of pre-existing veins in the porta hepatis and around biliary structures is seen as early as a few days after the onset of acute PVT. This is particularly conspicuous in the gallbladder wall, which is thickened and enhanced after contrast injection, and should be carefully distinguished from acute cholecystitis [28,59,60].

Disadvantages of CT include the use of ionizing radiation and potentially nephrotoxic contrast material which are particularly relevant in the pediatric population and in chronically affected patients, respectively. Moreover, CT provides a morphological examination of the abdominal anatomy, not giving information about the dynamic blood flow of the portal venous system, unlike CDUS examination [15]. Thus, CDUS and CT can be considered complementary.

#### 4.2.2. MR

MR is less operator dependent than CDUS, though more so than CT. Similar to CT, MR readily depicts typical findings of PVT including the extent of portal vein occlusion, potential occlusion of the superior mesenteric and splenic veins, and the extent of portosystemic collaterals and secondary changes in the liver (Figure 7). In the evaluation of portal cavernoma cholangiopathy, magnetic resonance cholangiopancreatography (MRCP) better depicts the changes in the biliary system than CT—avoiding ionizing radiation and providing high-quality images of the bile ducts and vessels—and replaces the diagnostic role of percutaneous transhepatic cholangiography and ERCP, reducing the risks of bleeding and acute pancreatitis associated with the latter [49]. One typical finding is the ‘‘wavy’’ appearance of the biliary ducts related to extrinsic compression by the collateral vessels, which was described in 87.5% of patients in one small retrospective study [46,47]. Other typical MR findings of the biliary system include areas of stenosis and dilatation, angulation of the common bile duct at the level of the superior pancreatic head (with an angle of ≤145 degrees considered significant) [61], and wall thickening of the gallbladder and bile ducts. Wall thickening may be related to intramural varices, as previously described, or ischemic fibrosis [46] (Figure 2). Epicholedochal collateral vessels may appear as punctate foci of intramural enhancement within the ducts on postcontrast MR images [61]. Bile duct and gallbladder stones are also well depicted on MRCP images. MR can also show other potential complications, including cholangitis and hepatic abscesses [46].

When the thrombus is located at the origin of the portal vein or extended to the splenic vein and/or the SMV, the intra- or peripancreatic cavernoma may form a pseudomass that encases the main bile duct and enhances during the portal phase after contrast injection. In such a case the misleading diagnosis of pancreatic tumor should be carefully avoided. MRCP coupled with MR angiography appears to be of great help in clarifying these differential diagnoses and avoiding the need for biopsy or fine-needle aspiration of a suspected mass, with higher sensitivity than CT [62].

MR imaging provides better soft tissue resolution and does not involve ionizing radiation; additionally, the paramagnetic contrast agents used in MR imaging have a better safety profile. Similar to CT, a technical issue of MR is its incapability to show dynamic blood flow of the portal venous system in clinically available protocols and its poor spatial resolution, especially in small children [63]; as previously mentioned for CT, CDUS imaging can be considered complementary. Major disadvantages are also that MR imaging is limited by longer acquisition times (30–60 min) and lower spatial resolution when compared with CT. Moreover, MR is considered quite invasive because it requires deep sedation or general anesthesia in younger children [15].

## 5. Role of Invasive Imaging Modalities in Portal Vein Thrombosis

### 5.1. Retrograde Portography

Retrograde portography is an interventional radiological procedure that provides visualization of the native intrahepatic portal venous system via retrograde opacification through wedge catheterization of the hepatic veins. Through the same technique, wedge hepatic pressure measurement, which reflects portal venous pressure, is possible, though in the case of PVT with cavernous transformation it is not reliable. When wedged hepatic venous pressure is within normal limits and intrasplenic pressure is significantly elevated, a presinusoidal nature of portal hypertension is assumed. Intravariceal pressure closely represents the portal and splenic pressure [2,64].

In children affected by extrahepatic PVT, the cure of portal hypertension with its associated complications can be achieved by surgical restoration of normal blood flow to the liver [65]. The preferred technique is the Meso-Rex shunt, which connects the SMV with the left intrahepatic portal branch close to the fetal insertion of the umbilical vein, called the Rex recess. However, a patent intrahepatic portal venous system is an indispensable prerequisite to perform Meso-Rex shunt surgery [63].

Retrograde portography is the gold standard imaging method to assess the intrahepatic portal venous system with high resolution: if patency of the native intrahepatic portal system is demonstrated, the surgical Meso-Rex bypass can be performed; vice versa, unnecessary operative explorations are avoided. A surgical spleno-renal (Warren) portosystemic shunt or a radiological transjugular intrahepatic portosystemic shunt (TIPS) can be proposed whenever intrahepatic vascular anatomy is compromised [66]. The choice between the two different approaches is based on other findings such as the presence of a prevalent cavernous vessel to land with the TIPS, the distance between the splenic and the left renal vein to make a surgical shunt, and local expertise/resource availability.

Conventional non-invasive imaging (CDUS, CT, MR) is not always reliable in assessing the patency of the native intrahepatic portal venous system and/or the Rex segment of the left portal vein in patients with cavernoma. At CDUS examination differentiating intrahepatic portal venous branches from venous collaterals in patients with the impaired portal venous flow can be challenging. Similar problems are faced by MR. One study compared MR and retrograde portography for the assessment of the Rex recess in children with extrahepatic PVT before Meso-Rex shunt surgery, demonstrating that only retrograde portography was reliable [63] (Figure 8).

Major concerns about transjugular retrograde portography in pediatric patients involve invasiveness, with radiation exposure and iodinated contrast agent administration [63].

Postoperative imaging assessment of the Meso-Rex shunt is treated in a dedicated paragraph.

### 5.2. Percutaneous Direct Portography

Percutaneous direct portography allows a fine assessment of the portal vein anatomy and PVT degree and extension; a functional and dynamic evaluation of the portal flow; and visualization of retrograde flow towards spontaneous portosystemic shunts [14]. The feasibility of recanalization of PVT is difficult to predict with preoperative imaging. Thus, the diagnostic role of percutaneous direct portography is emphasized at the beginning of a percutaneous revascularization procedure [67]. Endovascular transcatheter interventions are less invasive than surgery for the treatment of post-LT vascular complications, particularly in pediatric patients, and are considered the first-line approach for portal vein complications, with high rates of success shown by several studies [11,68].

Percutaneous transluminal angioplasty (PTA) relies on the dilatation of the occluded portal tract with balloon catheters during percutaneous transhepatic portography and it is highly effective for PVT [69] (Figure 9). In some cases of late-onset PVT, the fibrosis of the thrombosed segment may be an important determinant of recurrence and PTA may require additional metallic stent placement [8].

No consensus exists regarding the optimal indication for stent placement in PVT. Portal vein stent placement seems to be a safe and effective method for the treatment of post-transplantation PVT in pediatric recipients. The intermediate-term portal vein patency after stent placement is excellent [70] (Figure 10). In one study, after recanalization of the segment of portal vein affected by thrombosis through PTA, if an elastic recoil of more than 50% of the normal extrahepatic portal vein, a residual pressure gradient of more than 5 mmHg or a vascular dissection were found, stent placement was performed [8]. Several other investigators have reported a 100% portal vein patency rate after stent placement with various follow-up times [71,72]. These patency rates are superior to those previously reported with balloon angioplasty alone (27–50%) [73].

One argument against metallic stent deployment is that a metallic stent placed across an extrahepatic portal venous lesion may interfere with future portal venous anastomoses whenever retransplantation becomes necessary [74].

In the treatment of late-onset PVT with clinical signs of portal hypertension, decreased portal flow due to severe portosystemic shunt may contribute to recurrent PVT itself, and restoration of the portal flow with simultaneous embolization of portosystemic shunts may be important to achieve a sustained patency of the portal vein [8,14,24].

#### 5.2.1. Percutaneous Transhepatic Portography

The percutaneous transhepatic approach is the traditional method for portal vein catheterization and PVT treatment. The main advantage of direct transhepatic portography is that it allows accurate determination of the thrombus burden and extent, degree of stenosis, as well as direct portal venous pressure measurement which helps in the diagnosis of the cause of PVT [75].

#### 5.2.2. Percutaneous Transsplenic Portography

Percutaneous transsplenic portography represents a useful alternative access for endovascular portal vein treatments, especially when conventional transhepatic access is challenging or has failed.

A transsplenic access of the portal vein can be performed to achieve orthograde portography and to allow better conditions for recanalization in patients with tight portal vein occlusions and angled stenotic portal vein segments. Especially in young children with occluded, small, and short portal vein branches, this approach may help to achieve optimal conditions for portography and recanalization. The position and anatomy of the portal venous system in its intrahepatic and extrahepatic tracts varies after LT. Thus, recanalization via a transhepatic access can be unsuitable, and the antegrade transsplenic access adds valuable diagnostic information as well as an additional way to cross an unfavorable stenosis. Dilation of the venous splenic branches due to portal hypertension in these patients further increased the suitability of this approach.

The use of ultrasound guidance during puncture is mandatory to reduce the risk of complications [68,76]. Because of the high vascularity of the spleen, the major technical issue is represented by the risk of bleeding, which can be prevented with tract embolization during introducer sheath removal [77,78,79,80].

Several investigators reported that portal vein intervention via the transsplenic access is feasible and safe in LT recipients [77,78,79]. Some studies compared transhepatic and transsplenic access in terms of safety, and demonstrated that both methods are safe, with no difference in the rate of bleeding-related complications [78,79].

#### 5.2.3. Percutaneous Transmesenteric Portography

The use of a transmesenteric approach has been proposed when the other routes are not adequate to allow PVT treatment [80,81]. It can be either performed via direct image-guided percutaneous catheterization of the SMV or inferior mesenteric vein [81] or through SMV cannulation via a mini-laparotomy. This approach is more invasive than the other percutaneous routes, and it is less adopted [82]. 

A transmesenteric approach can be used both in early PVT and in chronic PVT when transhepatic and/or transsplenic access fails. Kim et al. and Cheng et al. described the use of the transmesenteric access for portal vein revascularization and stent placement for PVT during transplantation and early PVT [83,84]. Cavalcante et al. showed that PV revascularization using a transmesenteric approach may be an alternative procedure to reestablish the portal flow to the liver graft in selected cases, being a therapeutic addition to other treatment strategies currently used to treat chronic PVT. It has acceptable technical and clinical success rates, long-term patency, and a low rate of procedure-related complications [25] (Figure 11).

### 5.3. Transjugular Intrahepatic Portosystemic Shunt (TIPS)

Although the etiology of PVT in cirrhosis is incompletely understood, reduced portal blood flow velocity is thought to play a central role. Currently, anticoagulation is recommended as the first-line therapy for the management of PVT, though this may put patients with esophageal varices at increased risk for death from variceal hemorrhage.

Traditionally, PVT was considered a relative contraindication for TIPS placement. However, studies have shown that TIPS can facilitate improved blood flow, resulting in clot resorption and recanalization of PVT [85,86].

One study demonstrated that TIPS was effective for the treatment of PVT in patients with cirrhosis. Fifty-seven percent of cases had complete resolution of thrombosis, 30% had reduction in lumen obliteration and/or extension of thrombosis, and only 13% remained unchanged [86].

Moreover, a systematic review and meta-analysis exploring a total of 18 articles in which patients had a diagnosis of PVT and underwent TIPS for the management of PVT, showed that the use of TIPS may be a safe and feasible option in patients with underlying liver disease for the management of PVT, achieving portal patency in 86% of cases [87]. This study also showed that the use of TIPS may reduce clot burden with partial recanalization in 84% of patients and complete recanalization in 73% in patients with established PVT.

Acute thrombotic occlusions of the portal system can be managed by mechanical thrombectomy or pharmacologic thrombolysis. Chronic occlusions due to organized or refractory thrombus or fixed venous stenosis are better managed by balloon angioplasty and stent placement. Access into the portal venous system can be established through TIPS creation. A TIPS is also important in the setting of PVT associated with cirrhosis to decompress portal hypertension, improve portal venous flow, and prevent variceal hemorrhage [88]. The access to the portal system through a TIPS is also advisable in case of abundant perihepatic ascites or coagulopathy that preclude direct portal catheterization through percutaneous liver puncture. In the absence of portal hypertension or other contraindications such as ascites or coagulopathy, portal vein recanalization and mechanical thrombectomy can be also performed through direct transhepatic portal vein catheterization.

In patients with PVT without cirrhosis, the use of TIPS to improve portal flow is still controversial. If on the one hand TIPS may induce acceleration of the portal blood flow preventing the extension of thrombosis into the portal system, on the other hand, it raises concerns about long-term consequences of portal flow shunting on liver function [89].

Portal cavernoma is regarded as a relative contraindication to TIPS placement due to the high probability of technical failure. There are anecdotal reports of successful TIPS insertion in patients with a portal cavernoma in the absence of cirrhosis [89]. A study [90] suggested that TIPS creation by combination with a transhepatic or transsplenic approach is feasible and safe in the presence of portal cavernoma (Figure 12). However, TIPS was not recommended in the patients with a total fibrotic cord and without a large collateral vein or in those with extensive SMV thrombosis. By definition, TIPS can decrease the portosystemic pressure gradient, thus reducing the incidence of variceal bleeding [90]. In patients with jaundice or recurrent biliary symptoms due to portal cholangiopathy, although feasible in a limited number of patients, TIPS has been primarily used to address the biliary complications, allowing portal decompression, and staged biliary surgery [28] (Figure 13).

## 6. Post-Surgical Assessment of Meso-Rex Shunt

Nowadays, the Meso-Rex bypass is the treatment of choice for extrahepatic PVT in pediatric patients, given its unique ability to re-establish physiological hepatic portal venous blood flow [66,91].

CDUS is the first-line modality both in the early postoperative period and for regular follow-up. An interesting study showed that during the first week the flow rate increased rapidly within the bypass and intrahepatic portal branches to reach values twice or three times higher than those intraoperatively seen. This was mirrored by a growth in the diameter of the intrahepatic portal veins [66]. 

CT or MR can be used to depict Meso-Rex bypass anatomy in complex cases where CDUS is inconclusive. Multiplanar and 3-D reconstructions are helpful to illustrate the course and patency of the bypass and the SMV, and the multiple anatomical variants of shunt anatomy that can be observed [66,92].

Stenosis and thrombosis of the bypass graft typically occur weeks to years after surgery and may be clinically evident with recurrence of portal hypertension and impaired liver function tests. Thrombosis or high-grade stenosis represents the main cause of bypass graft occlusion. CT may show a low-density cord, representing the occluded bypass. CDUS can show the thrombus with a hypoechoic appearance associated with an absence of portal flow. In these situations, stenosis and thrombosis can be treated with transhepatic, transsplenic, or transmesenteric portography and angioplasty. Balloon dilation can be performed when the transtenotic pressure gradient is greater than 5 mmHg or when a stenosis is radiologically confirmed [66] (Figure 14).

## 7. Discussion

PVT is relatively uncommon in the general population, but it is relatively frequent among cirrhotic patients, representing a predictable event in the natural evolution of liver diseases. Local or systemic pro-thrombotic factors, alone or together with other predisposing factors, can play an important role in PVT pathogenesis. The consequent changes in hepatic and splanchnic hemodynamic are responsible for a mild impairment in liver function, also in the absence of an overt liver disease, or can precipitate a pre existing metastable clinical status in cirrhotic patients. Moreover, PVT might have indirect effects on other abdominal organs, causing intestinal ischemia and infarction, or predisposing to vascular neoformation and gastrointestinal bleeding. The identification of the broad spectrum of manifestations of PVT is essential to provide a prompt diagnosis and treatment via interventional radiology when feasible, to preserve patient health and life.

The history of PVT has been characterized by difficulties in diagnosis and treatment, which, today, have almost been overcome. Recent protocols for periodic CDUS and CT have led to early diagnosis of subacute or late-onset PVT prior to the development of clinical symptoms or complete portal vein obstruction in patients affected by cirrhosis.

Imaging plays an essential role in the evaluation of PVT and its related manifestations, especially in patients undergoing LT. It may prevent regrafting and patient loss. In both adult and pediatric liver transplant recipients, technical advances in diagnostic and interventional radiology have contributed to significantly improving grafts’ and consequently patients’ survival rates.

Although interventional radiological techniques have made a huge step forward in recent years, the role of percutaneous revascularization in chronic extrahepatic PVT has not been established yet, especially in pediatric patients with long-term fibrotic occlusion [37]. Few studies have reported successful pediatric percutaneous portal vein recanalization, suggesting that this procedure can be considered a feasible alternative to the standard surgical option [7,8,65,93]. Portal vein recanalization (PVR) has been also proven to be effective in the PVT treatment of both cirrhotic and non-cirrhotic patients [94]. During PVR, an aspect to consider is that decreased portal flow due to spontaneous portosystemic shunt may contribute to recurrent PVT, and restoration of the portal flow with simultaneous embolization of portosystemic shunts may be important to achieve a sustained patency of the revascularized portal vein [8,14].

Nowadays, in pediatric patients, the meso-rex bypass is still the treatment of choice for extrahepatic PVT, given its ability to re-establish physiological hepatic portal venous blood flow [66]. Indeed, it has been recommended by the Baveno consensus [95] that meso-Rex bypass should be considered in all children with complications of portal cavernoma in the early stage of extrahepatic PVT. Meso-rex bypass is considered the most physiological shunting and the only “curative surgery” [95]. Nevertheless, interventional radiological revascularization techniques have the potential to restore the physiological flow to the liver, through the native portal vein, with less invasiveness compared to surgery. However, the feasibility of PVR is often not predictable using preoperative imaging. In a study [67], preoperative imaging lacked diagnostic accuracy in predicting when PVR was not feasible in chronic PVT. Portography performed at the beginning of the procedure seems mandatory for deciding whether PVR should be attempted.

Another important issue regards the choice between PVR alone or associated with TIPS. As proposed by an interesting study [67], as the aim of PVR is to alleviate portal hypertension, the combined approach should be taken into account. Moreover, TIPS would be likely useful when portal hypertension is related to an intrahepatic obstruction to the portal circulation. As intrahepatic resistances are normal in most patients with portal vein obstruction, the benefit of associating TIPS with PVR remains unclear and should be evaluated on an individual basis.

PVT treatment is challenging, and the patient’s management is often decided on a case-by-case basis. Anticoagulation has been proposed in cirrhotic or non-cirrhotic patients who are predisposed to venous thrombosis. In particular, the European Association for the Study of the Liver guidelines recommended considering therapeutic anticoagulation in patients with cirrhosis and PVT [38]. Despite these guidelines, the efficacy and safety of anticoagulation in this population are uncertain. A higher tendency to bleeding in cirrhotic patients and the frequent presence of gastroesophageal varices are considered a relative contraindications to anticoagulation, making this therapy a hard decision [96].

In conclusion, non-invasive multimodality imaging is essential for a panoramic characterization of PVT and its associated alterations. Invasive techniques offer a better understanding of portal circulation hemodynamics and provide the route for percutaneous revascularization, which should be more extensively considered in the future for the management of acute and chronic PVT.

## Figures and Tables

**Figure 1 diagnostics-12-02628-f001:**
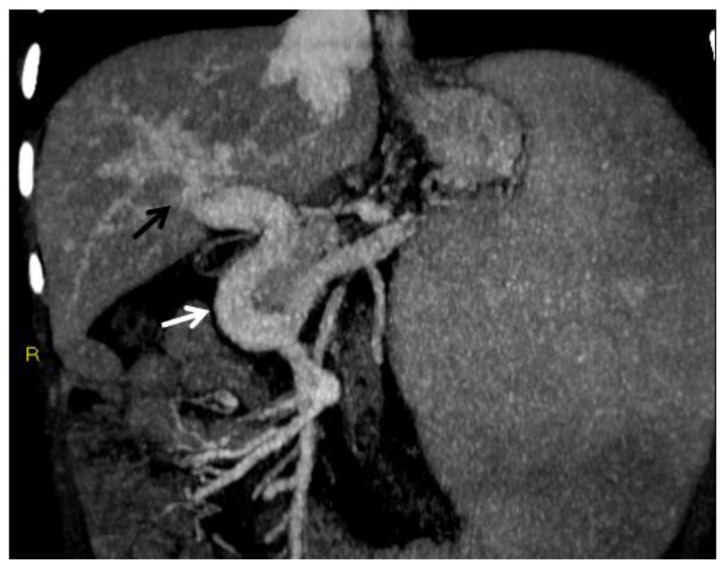
CT of an 8-year-old female who presented with hematemesis due to variceal bleeding. Portal phase, maximum intensity projection CT image reformatted on the coronal plane shows a large, tortuous collateral vein of the extrahepatic cavernoma at the porta hepatis (white arrow). Cavernous transformation of the intrahepatic portal branches is also seen (black arrow).

**Figure 2 diagnostics-12-02628-f002:**
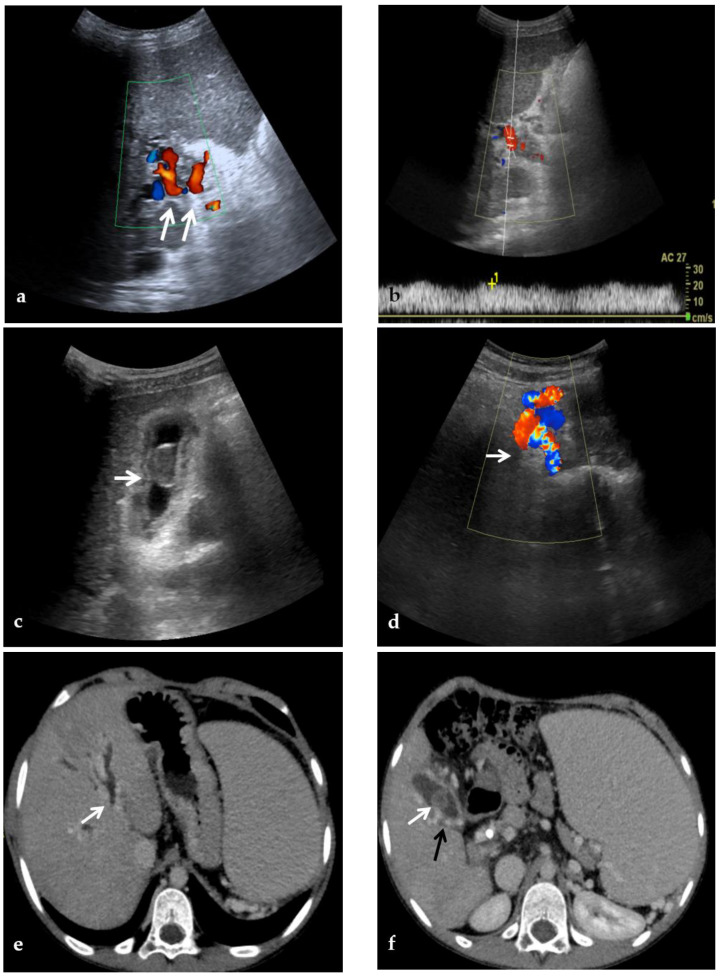
CDUS, CT, and MR images of a 9-year-old male with intrahepatic and extrahepatic cavernous transformation of the portal vein and portal biliopathy who presented with jaundice and cholangitis. (**a**) A Color-Doppler Ultrasound image shows multiple collateral vessels at the porta hepatis (arrows) and the absence of the main portal trunk. (**b**) Spectral evaluation of the major collateral vein of the cavernoma shows an hepatopetal flow with a velocity within normal ranges, which could potentially be misleading by an unexperienced operator. (**c**) An ultrasound image scan through the gallbladder shows a mildly echogenic endoluminal stone in the gallbladder (arrow). (**d**) Color-Doppler Ultrasound image shows multiple dilated collateral cavernous vessels in the gallbladder fossa (arrow). (**e**) Portal phase axial CT image shows dilation of intrahepatic biliary ducts (arrow) due to extrinsic compression of the common biliary duct by the cavernoma. (**f**) A caudal axial CT view in the portal venous phase shows a mildly hyperdense stone in the gallbladder (white arrow) which is surrounded by dilated cavernous vessels (black arrow). (**g**) Axial T2-weighted MR image shows intrahepatic biliary dilatation (arrows) due to common biliary duct extrinsic obstruction by prominent paracholedochal collateral cavernous vessels. (**h**) A caudal axial T2-weighted MR view shows wall thickening of the gallbladder (white arrows) which contains other multiple small gallstones in the fundus (black arrow).

**Figure 3 diagnostics-12-02628-f003:**
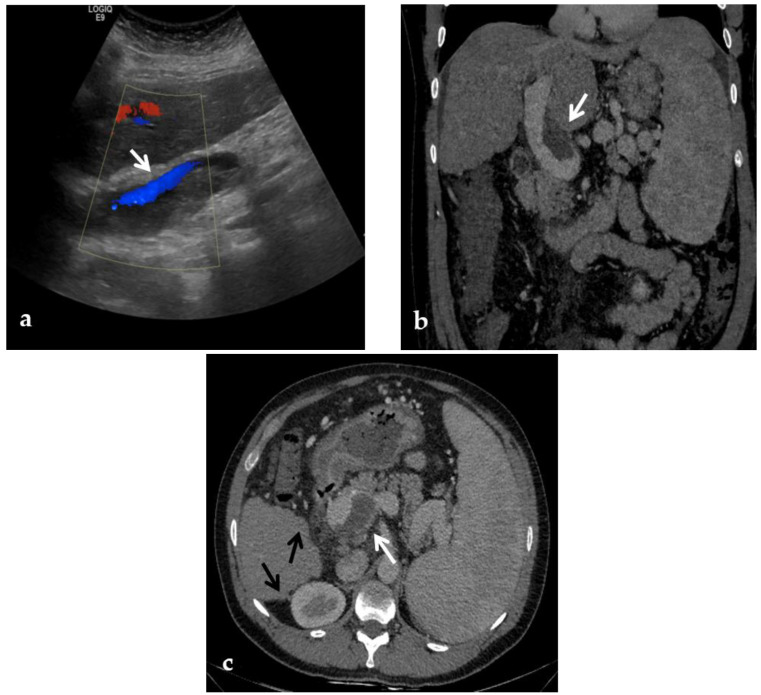
Color-Doppler Ultrasound and CT images of portal vein thrombosis in a 54-year-old male suffering from liver cirrhosis. (**a**) Color-Doppler Ultrasound image shows a narrow hepatopetal flow (arrow) despite a large endoluminal thrombus in the extrahepatic portal trunk. (**b**) Portal phase coronal CT image shows the lack of luminal enhancement and increased portal vein caliber due to a large endoluminal thrombus in the extrahepatic trunk (arrow). (**c**) Portal phase axial CT image shows the extension of the portal vein thrombosis up to the splenomesenteric confluence (white arrow); liver cirrhosis is visible on the the liver surface that presents lobulated margins (black arrows).

**Figure 4 diagnostics-12-02628-f004:**
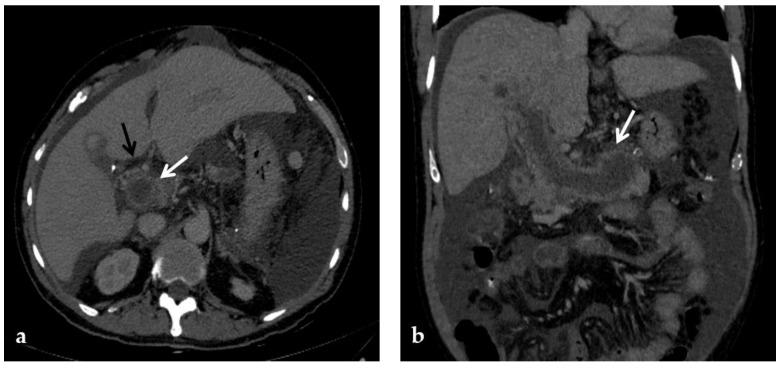
CT images of portal vein thrombosis in a 62-year-old male affected by myelofibrosis, who presented with intense abdominal pain. (**a**) Portal phase axial CT image shows a lack of luminal enhancement in the porta hepatis, increased portal vein caliber with enhancing walls, and a low-attenuating thrombus within the lumen, consistent with acute portal vein thrombosis (white arrow). Multiple collateral vessels forming a cavernoma are seen at the hepatic hilum (black arrow). (**b**) Portal phase coronal CT image better shows the extension of portal vein thrombosis, also involving the splenic vein (arrow).

**Figure 5 diagnostics-12-02628-f005:**
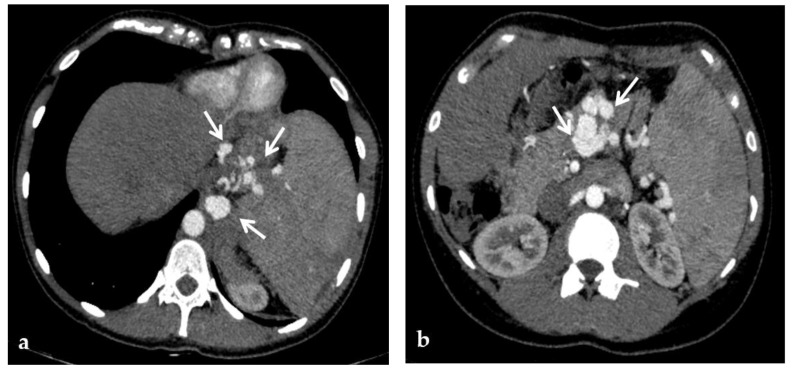
CT of a 21-year-old female who underwent deceased-donor orthotopic liver transplantation (split II-III) for biliary atresia, complicated by the development of portal vein thrombosis and cavernous transformation, who presented with melena and hematemesis 12 years after transplantation. (**a**) Arterial phase axial CT images show multiple arterial aneurysms of the diaphragmatic and gastric arteries, with hypertrophic submucosal vessels in the gastric fundus (arrows). (**b**) Arterial phase axial CT image also shows multiple large arterial aneurysms of the splenic artery (arrows).

**Figure 6 diagnostics-12-02628-f006:**
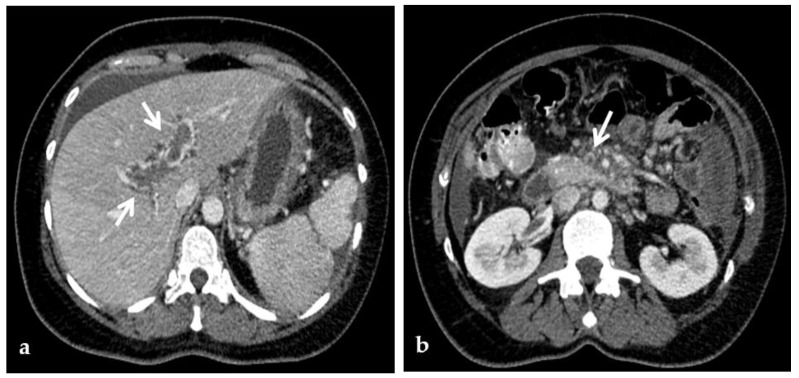
CT of a 39-year-old female with no previous relevant clinical history, who presented with an intense acute abdominal pain. (**a**) Portal phase axial CT image shows acute massive thrombosis of left and right intrahepatic portal branches (arrows). Portal phase axial (**b**) and coronal (**c**) CT images show the extension of the thrombus to the superior mesenteric vein (arrow). No signs of intestinal infarction were present, and the patient was submitted to transjugular intrahepatic portosystemic shunt placement and mechanical thrombectomy.

**Figure 7 diagnostics-12-02628-f007:**
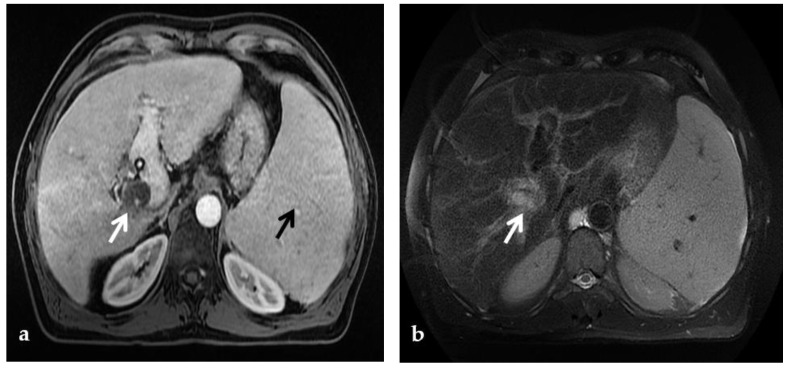
MR of a 50-year-old male with chronic portal vein thrombosis and portal hypertension. (**a**) Axial T1-weighted contrast-enhanced MR image shows low signal intensity within the right branch of the portal vein (arrow). Splenomegaly is a typical hallmark of portal hypertension (black arrow). (**b**) Axial T2-weighted MR image shows that portal vein thrombosis appears of increased signal intensity (arrow).

**Figure 8 diagnostics-12-02628-f008:**
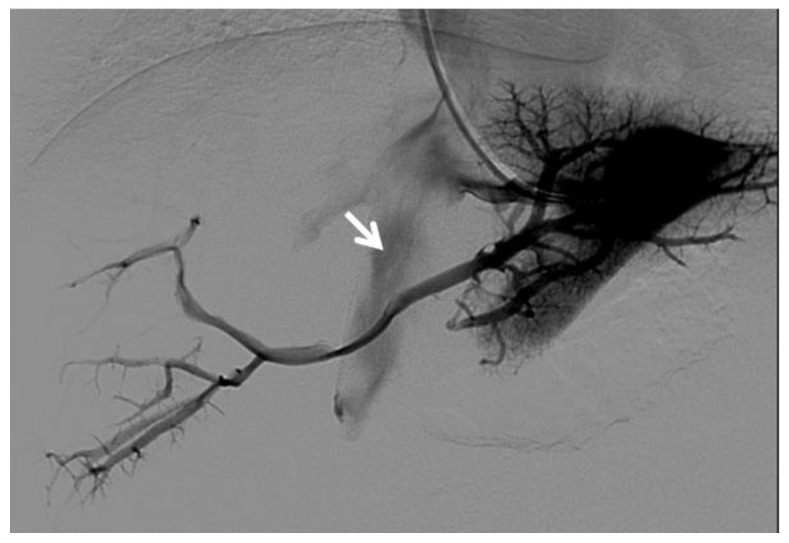
Right oblique anterior (30° cranial) view of a transjugular retrograde portography performed in a 9-year-old female with extrahepatic portal vein cavernoma due to a neonatal umbilical vein catheterization and portal vein thrombosis, who complained of symptoms of portal hypertension. She was a candidate for Meso-Rex bypass surgery. The image shows retrograde filling of the intrahepatic portal vessels with the complete visualization of the Rex segment (arrow) after wedged left hepatic venography. Meso-Rex bypass was then surgically performed.

**Figure 9 diagnostics-12-02628-f009:**
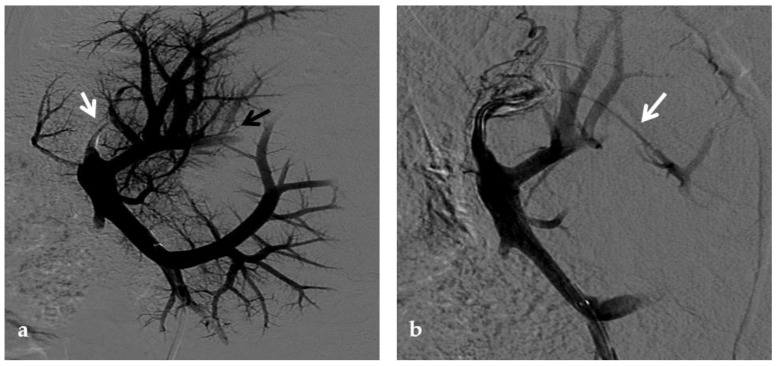
Anteroposterior view of a percutaneous transhepatic portography imaging series performed for chronic complete portal vein thrombosis with cavernous transformation in an 11-year-old boy with a history of gastrointestinal bleeding and portal hypertension 10 years after deceased-donor orthotopic liver transplantation (split II–III) for biliary atresia. (**a**) Portography image shows a regular representation of the segment 3 portal branch, while the segment 2 portal branch has a peripheral defect of opacification due to flow reversal, resulting from portal cavernoma and extrahepatic capsular feeding from gastric vessels (black arrow). Complete occlusion of the extrahepatic portal vein at the hepatic hilum (white arrow) is demonstrated. (**b**) Portography image shows the opacification of a very tiny vascular structure (arrow) which represents what remains of the thrombosed extrahepatic portal trunk. (**c**) Fluoroscopy image shows angioplasty with a non-compliant 8-mm balloon catheter performed on a stiff guidewire which was advanced across the obstructed tract. (**d**) Control portography shows a re-expanded extrahepatic portal vein (arrow) with normal spleno-portal opacification. Portal vein thrombosis recurred one year after treatment and required repeated angioplasty and metal stenting (shown in the next figure).

**Figure 10 diagnostics-12-02628-f010:**
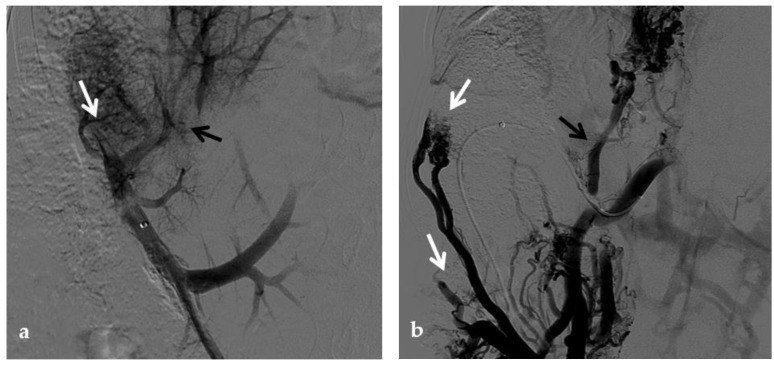
Anteroposterior view of a percutaneous transhepatic portography imaging series in the same patient as in Figure 9. (**a**) Portography image shows recurrence of complete extrahepatic portal vein thrombosis (white arrow) 1 year after the first angioplasty treatment. Flow reversal in the segment 2 portal branch resulting from portal cavernoma is still evident (black arrow). (**b**) After transhepatic portal vein recanalization with the angiographic catheter, the splenic portogram shows jejunal cavernoma (white arrows) and gastric (black arrow) varices, due to complete obstruction of the extrahepatic portal trunk. Embolization of varices was performed using a mixture of N-butyl cyanoacrylate and iodized oil and metallic coils. (**c**) Portography image after angioplasty of the extrahepatic portal vein shows successful endoluminal filling of the portal vein, with a restored but still irregular profile (white arrow). Note subtraction artifacts (black arrows) representing lipiodol and coils in the cavernoma and varices. (**d**) Final portogram performed after deployment of a 10-mm self-expandable bare metal stent to treat residual stenosis of the extrahepatic portal vein; the main portal vein is now regularly opacified with adequate size and both intrahepatic portal branches present hepatopetal flow.

**Figure 11 diagnostics-12-02628-f011:**
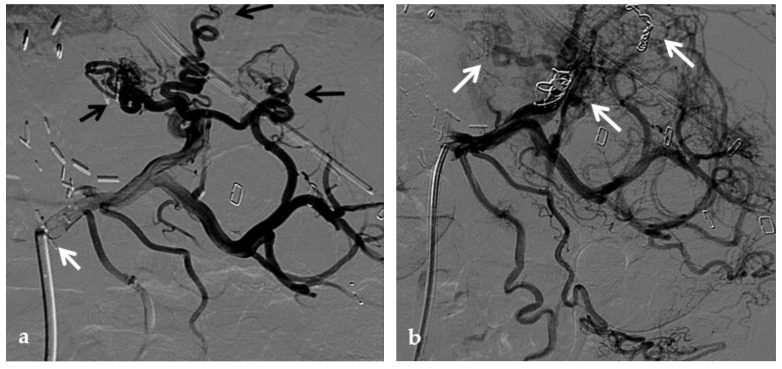
Anteroposterior view of a percutaneous transmesenteric portography imaging series of an acute complete portal vein thrombosis occurred early after pediatric split-liver transplantation in a 1-year-old female with biliary atresia. (**a**) Superior mesenteric portography performed through a surgically-placed transmesenteric sheath (white arrow) shows total occlusion of the main portal vein and its intrahepatic branches, with opacification of several portosystemic varices (black arrows). (**b**) Superior mesenteric portography shows effective embolization of the portosystemic shunts to reduce the flow-steal phenomenon. A mixture of N-butyl cyanoacrylate and iodized oil and metallic microcoils (arrows) were used. (**c**) Portography image shows angioplasty with a 5-mm non-compliant balloon catheter performed on a stiff guidewire advanced through the common portal vein (white arrow). Note the spontaneous remaining opacification caused by the mixture of N-butyl cyanoacrylate and iodized oil (black arrows). (**d**) Final superior mesenteric portography shows restored opacification of the portal vein anastomosis, the umbilical recess of the portal vein (black arrow) and the intrahepatic segment 2 and segment 3 branches (white arrows), with no contrast filling of the portosystemic shunts.

**Figure 12 diagnostics-12-02628-f012:**
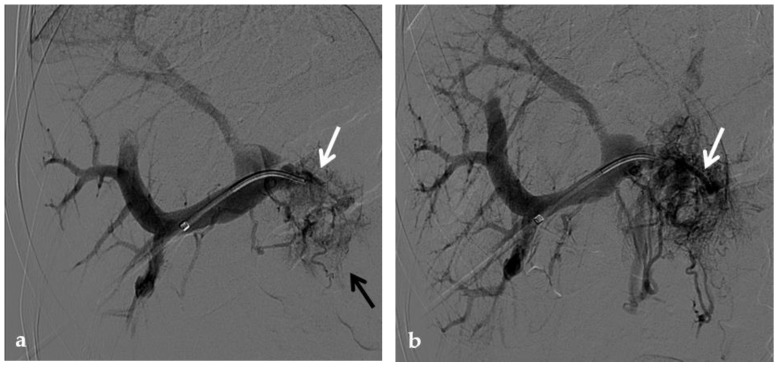
Anteroposterior view of a percutaneous transhepatic and transsplenic portography imaging series of a chronic complete extrahepatic portal vein thrombosis with portal cavernous transformation in a 17-year-old boy with a history of gastrointestinal bleeding and portal hypertension after liver transplant for biliary atresia. (**a**) Percutaneous transhepatic portography shows complete extrahepatic portal vein thrombosis (white arrow), with opacification of an irregular cavernomatous network (black arrow). (**b**) Opacification of a very tiny fibrotic cord representing the thrombosed portal trunk is seen (arrow). Several attempts to cross the portal vein occlusion failed. (**c**) Percutaneous transsplenic anterograde portography shows complete extrahepatic portal vein thrombosis and cavernous transformation (arrow) from a different perspective. (**d**) The tiny residual of the thrombosed main portal trunk (arrow) was easily recognized on the transsplenic superior mesenteric vein portography. (**e**) Fluoroscopic image shows angioplasty of the main portal vein, performed through a 12-mm non-compliant balloon catheter. The focal notch (arrow) represents the tight anastomotic stenosis that probably led to secondary thrombosis. (**f**) Fluoroscopic image shows the resolution of the focal notch (arrow) after high-pressure inflation of the non-compliant 12-mm balloon catheter. (**g**) Control portography shows an expanded main portal vein with improved hepatopetal portal flow, but still with irregular profiles (arrow). (**h**) Through a hybrid transhepatic and transjugular approach under combined ultrasound and fluoroscopic guidance, the connection between the intrahepatic portal vein and the vena cava was created advancing a 0.018” micro guidewire into the right atrium (arrow). The wire was snared through the right transjugular access providing a through-and-through access for precise transjugular intrahepatic portosystemic shunt (TIPS) placement. The completely image-guided hybrid approach allowed to reduce the attempts of transjugular intrahepatic portal vein puncture. (**i**) Portography image shows the deployment of two imbricated Viatorr stents to create the TIPS (arrows), with regular intrastent opacification. Despite previous angioplasty, a prestenotic aspect of the main portal vein is seen (black arrow). (**j**) Portography image shows a self-expandable 9-mm metallic stent placed to cover the main portal vein trunk stenosis (white arrow). Control portography shows regular mesenteric-portal vein opacification with TIPS patency and preserved opacification of intrahepatic portal branches (black arrows); cavernous vessels disappeared.

**Figure 13 diagnostics-12-02628-f013:**
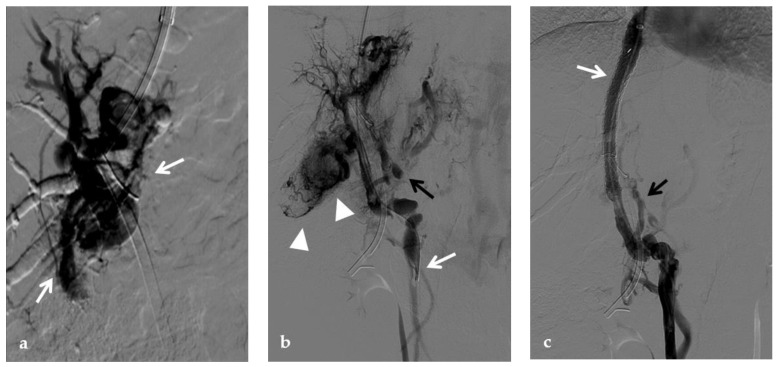
Anteroposterior view of a transjugular intrahepatic portosystemic shunt portography imaging series in the same patient of Figure 2. (**a**) Transjugular intrahepatic portal puncture confirms the chronic occlusion of the main portal vein with intrahepatic cavernous transformation (arrows). (**b**) After transjugular catheterization of the main cavernous vessel portography shows a patent superior mesenteric vein (white arrow) with other minor cavernous vessels at the hepatic hilum (black arrow). Note the aberrant paracholecystic veins forming portoportal collaterals to bypass the obstructed segment of the portal vein (arrowheads). (**c**) Portography performed after transjugular intrahepatic portosystemic shunt placement shows regular stent opacification (white arrow), with reduced visualization of the cavernoma (black arrow) and disappearance of paracholecystic veins.

**Figure 14 diagnostics-12-02628-f014:**
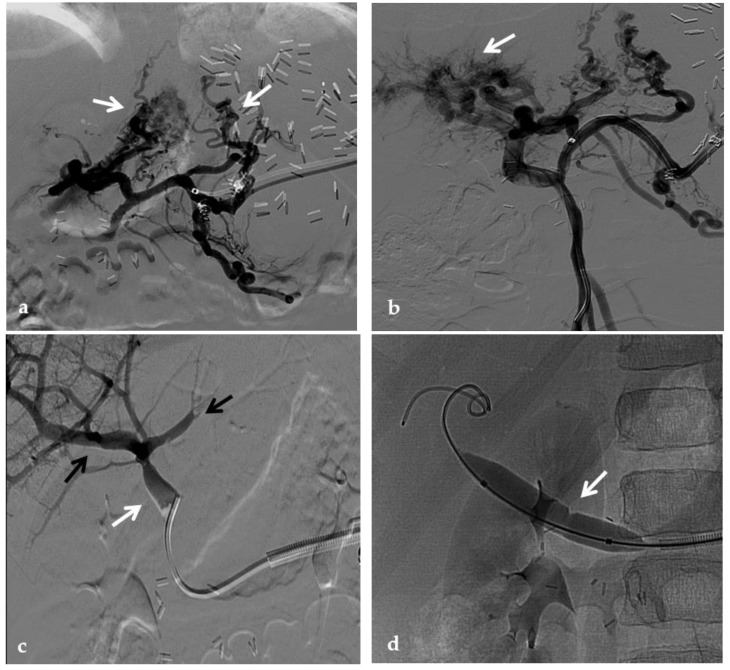
Anteroposterior view of a percutaneous transsplenic portography imaging series of an occluded Meso-Rex bypass in the same patient of Figure 8. (**a**) Percutaneous transsplenic portography shows extensive jejunal and gastric varices (arrows), with no opacification of the Meso-Rex bypass. (**b**) After superior mesenteric vein catheterization, portography shows the portal cavernoma (arrow) with no opacification of the native intrahepatic portal branches (whose patency was documented with retrograde portography in Figure 8). (**c**) Portography image shows successful recanalization of the mesenteric anastomosis of the Meso-Rex bypass (white arrow), with intrahepatic portal branches opacification (black arrows). (**d**) X-Ray image shows angioplasty of the bypass performed through a 9-mm non-compliant balloon catheter. During angioplasty, a focal notch (arrow) representing the mesenteric anastomotic stenosis is clearly visible. (**e**) Portography image acquired after bypass recanalization and angioplasty shows regular opacification of the intrahepatic portal system (arrows). (**f**) Portography image from the superior mesenteric vein shows a re-expanded Meso-Rex bypass with complete opacification in the mesenteric anastomotic tract (arrow). The intrahepatic portal system is regularly represented; the portal cavernoma and gastro-jejunal varices disappeared.

## Data Availability

Not applicable.

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
