# Peer review of "The Role of Imaging in Portal Vein Thrombosis: From the Diagnosis to the Interventional Radiological Management"

_diagnostics, 2022, doi:10.3390/diagnostics12112628_

Round 1

Reviewer 1 Report

Paper It Is well wttwn and interesting however i have few minor criticisms

Authors should produce a table with etiopathogenesis of pvt related findings

Particularly they Simply to produce data in disease related to PVT

Discussioni could be improved in terms of lenght and message to readers

Author Response

Reviewer's comment:

-Paper It Is well wttwn and interesting however i have few minor criticisms

-Authors should produce a table with etiopathogenesis of pvt related findings

Particularly they Simply to produce data in disease related to PVT

Thank you for this important suggestion. A table was added.

Discussion could be improved in terms of lenght and message to readers

Thank you for this insightful comment. The conclusion section was renamed discussion and added with the following paragraphs with some added references:

"Although interventional radiological techniques have made a huge step forward in recent years, the role of percutaneous revascularization in chronic extrahepatic PVT has not been established yet, especially in pediatric patients with long-term fibrotic occlusion [37]. Few studies have reported successful pediatric percutaneous portal vein recanalization, suggesting that this procedure can be considered a feasible alternative to the standard surgical option [7,8,65,93]. PVR has been also proven to be effective in PVT treatment of both cirrhotic and non-cirrhotic patients [94]. During PVR, an aspect to consider is that decreased portal flow due to spontaneous portosystemic shunt may contribute to recurrent PVT, and restoration of the portal flow with simultaneous embolization of portosystemic shunts may be important to achieve a sustained patency of the revascularized portal vein [8,14].

Nowadays, in pediatric patients the meso-rex bypass is still the treatment of choice for extrahepatic PVT, given its ability to re-establish physiological hepatic portal venous blood flow [95]. Indeed, it has been recommended by the Baveno consensus [96] that meso-Rex bypass should be considered in all children with complications of portal cavernoma in the early stage of extrahepatic PVT. Meso-rex bypass is considered the most physiological shunting and the only “curative surgery” [96]. Nevertheless, interventional radiological revascularization techniques have the potential to restore the physiological flow to the liver, through the native portal vein, with less invasiveness compared to surgery. However, feasibility of PVR is often not predictable using preoperative imaging. In a study [67], preoperative imaging lacked diagnostic accuracy in predicting when PVR was not feasible in chronic PVT. Portography performed at the beginning of the procedure seems mandatory for deciding whether PVR should be attempted.

Another important issue regards the choice between PVR alone or associated with TIPS. As proposed by an interesting study [67] ,as the aim of PVR is to alleviate portal hypertension, the combined approach should be taken into account. Moreover, TIPS would be likely useful when portal hypertension is related to an intrahepatic obstruction to the portal circulation. As intrahepatic resistances are normal in most patients with portal vein obstruction, the benefit of associating TIPS to PVR remains unclear and should be evaluated on an individual basis.

PVT treatment is challenging, and the patient’s management is often decided on a case-by-case basis. Anticoagulation has been proposed in cirrhotic or non-cirrhotic patients who are predisposed to venous thrombosis. In particular, the European Association for the Study of the Liver guidelines recommended to consider therapeutic anticoagulation in patients with cirrhosis and PVT [38]. Despite these guidelines, the efficacy and safety of anticoagulation in this population is uncertain. A higher tendency to bleeding in cirrhotic patients and the frequent presence of gastroesophageal varices are considered a relative contraindication to anticoagulation, making this therapy a hard decision [97].

In conclusion, non-invasive multimodality imaging is essential for a panoramic characterization of PVT and its associated alterations. Invasive techniques offer a better understanding of portal circulation hemodynamics and provide the route for percutaneous revascularization, which should be more extensively considered in the future for the management of acute and chronic PVT."

Reviewer 2 Report

The work of Dr. Marra and the team provides a comprehensive and extensive review of portal vein thrombosis. I recommend publishing the review in its current form. 

Author Response

Reviewer's comment:

The work of Dr. Marra and the team provides a comprehensive and extensive review of portal vein thrombosis. I recommend publishing the review in its current form. 

We thank the reviewer for the positive comments and the time dedicated to the review.

Reviewer 3 Report

This paper is interesting and useful. The authors illustrated diagnostic and interventional imaging for the characterization and treatment of portal vein thrombosis (PVT). The broad spectrum of congenital and acquired PVT manifestations is illustrated, with focus on the pediatric population; diagnostic and interventional imaging techniques are described. This review article is of great importance for researchers in this field. For researchers and readers, it will be a good reference for the diagnosis and treatment of PVT. Therefore, the manuscript is suitable for publication.

Author Response

Reviewer's comment:

This paper is interesting and useful. The authors illustrated diagnostic and interventional imaging for the characterization and treatment of portal vein thrombosis (PVT). The broad spectrum of congenital and acquired PVT manifestations is illustrated, with focus on the pediatric population; diagnostic and interventional imaging techniques are described. This review article is of great importance for researchers in this field. For researchers and readers, it will be a good reference for the diagnosis and treatment of PVT. Therefore, the manuscript is suitable for publication.

We thank the reviewer for the positive comments and the time dedicated to the review.